Comparative survival analyses among captive chimpanzees (Pan troglodytes) in America and Japan

http://orcid.org/0000-0002-9118-9202 Che-Castaldo Judy 1
Havercamp Kristin 2
Watanuki Koshiro 2
http://orcid.org/0000-0002-8147-2725 Matsuzawa Tetsuro 3 4
http://orcid.org/0000-0002-1026-6270 Hirata Satoshi 2 5 hirata.satoshi.8z@kyoto-u.ac.jp
http://orcid.org/0000-0002-1819-4136 Ross Stephen R. 6 sross@lpzoo.org
1 Alexander Center for Applied Population Biology, Conservation & Science Department, Lincoln Park Zoo , Chicago, Illinois , United States
2 Wildlife Research Center, Kyoto University , Kyoto , Japan
3 Chubu Gakuin University , Gifu , Japan
4 Division of the Humanities and Social Science, California Institute of Technology , Pasadena, California , United States
5 Kumamoto Sanctuary, Kyoto University , Kumamoto , Japan
6 Lester E. Fisher Center for the Study and Conservation of Apes, Lincoln Park Zoo , Chicago, Illinois , United States
Gillespie Joseph
Electronic publication date: 2021 Aug 12
Publication date: 2021
Volume: 9
Electronic Location ID: e11913
Received 2021 Mar 26; Accepted 2021 Jul 14
Copyright: © 2021 Che-Castaldo et al.
Copyright year: 2021
Copyright holder: Che-Castaldo et al.
License: This is an open access article distributed under the terms of the Creative Commons Attribution License, which permits unrestricted use, distribution, reproduction and adaptation in any medium and for any purpose provided that it is properly attributed. For attribution, the original author(s), title, publication source (PeerJ) and either DOI or URL of the article must be cited.
License URL: https://creativecommons.org/licenses/by/4.0/

Keywords: Chimpanzee, Survival analyses, Life table, Longevity, Mortality, Life history, America, Japan, Captivity

Funding: SGU MEXT and JAWK International Scholarship to Kristin Havercamp JSPS 18H05524 to Satoshi Hirata 18H05524 CCSN-R2 to Tetsuro Matsuzawa Lincoln Park Zoo Women’s Board support to Stephen R. Ross This research was supported by the Super Global University Ministry of Education, Culture, Sports, Science and Technology of Japan (SGU MEXT) and JAWK International Scholarship to Kristin Havercamp, the Japan Society for the Promotion of Science (No. 18H05524) to Satoshi Hirata, the Core-to-Core program CCSN-R2 to Tetsuro Matsuzawa and the Leading Graduate Program of Primatology and Wildlife Science (PWS-U04) to Kristin Havercamp, Satoshi Hirata and Tetsuro Matsuzawa. Data collection of chimpanzees in Japan was supported by the Great Ape Information Network (GAIN). Support for Stephen Ross was provided by the Lincoln Park Zoo Women’s Board. The funders had no role in study design, data collection and analysis, decision to publish, or preparation of the manuscript.

==============================
Detailed, long-term datasets on the life histories of long-lived species such as great apes are necessary to understand their survival patterns but are relatively rare. Such information requires prolonged and consistent record-keeping over many generations, so for chimpanzees (Pan troglodytes), this equates to many decades of input. As life history variables can be altered by differences in environmental influences (whether natural or artificial), there is substantial value to being able to compare across populations. Here, we present the first comparative analysis of life history data for two ex situ chimpanzee populations residing in North America (1975–2020; n = 730) and Japan (1980–2020; n = 660). Overall, survival patterns were similar between regions, and the median life expectancy from birth is estimated at 35.7 (95% CI = [32.4–40.0]) years for females and 30.1 (27.3–34.3) years for males across both populations. Females who survive to their first birthday are estimated to survive 42.4 (40.0–46.3) years and males 35.5 (32.6–38.0) years. We found that birth type (wild-born or captive-born) did not influence survival patterns in either population, but there were differential effects of sex on longevity. In the America population, males had higher mortality rates than females, whereas in the Japan population we found no differences between the sexes. First year mortality did not differ between populations for males (18–20%), but for females it was lower in America (15%) compared to Japan (25%). Survival patterns of chimpanzees in the present study will be useful for future investigation into potential causes of regional differences and cross-species comparisons.

Introduction

Studying life history patterns of non-human primates is important for understanding the evolution of human life histories and senescence (Bronikowski et al., 2011). In particular, chimpanzee (Pan troglodytes) survival and mortality patterns have been studied to explore differences with other species, among populations of chimpanzees in the wild, and between wild and captive populations (e.g., Earnhardt et al., 2003; Hill et al., 2001; Thompson et al., 2007; Muller & Wrangham, 2014; Tidière et al., 2016; Wood et al., 2017; Davison & Gurven, 2021). Detailed, long-term datasets on life histories are rare but important for capturing survival rates at later ages, especially for long-lived species such as chimpanzees. Comparing chimpanzee survival across different captive populations may expose similarities or differences in survival patterns which could be further examined for determining best management strategies such as husbandry techniques, diet and veterinary care.

Chimpanzees are one of humans’ closest living relatives and have been studied for decades, which has allowed for the comparison of life history patterns to shed light on human evolution (e.g., Hawkes, Smith & Robson, 2009; Hill et al., 2001; Muller & Wrangham, 2014; Thompson et al., 2007; Wood et al., 2017; Davison & Gurven, 2021). Life expectancy at birth for hunter-gatherers ranges from 21–37 years across groups and between 26–43% of people survive to age 45 (Gurven & Kaplan, 2007), similar to what has been found in a wild (Wood et al., 2017) and a captive chimpanzee population (Havercamp et al., 2019). The average life expectancy of wild chimpanzees varies across populations (Table 1). Data from five sites across Africa show that individuals who survive to adulthood (14 y) have a life expectancy of approximately 29 years (Hill et al., 2001). In Kanyawara in East Africa, adult chimpanzees (14 y) can expect to live 38 years (Muller & Wrangham, 2014) and at Ngogo, also in East Africa, 43 years (Wood et al., 2017). In captivity in Japan, a 14-year old individual is expected to reach 42 years of age (Havercamp et al., 2019). Longevity estimates of chimpanzees from birth are lower due to the high risk infant period, from 13 years across five sites (Hill et al., 2001), 19 years at Kanyawara (Muller & Wrangham, 2014), 28 years in captivity (Havercamp et al., 2019), to 33 years at Ngogo (Wood et al., 2017). Typically, mammals live longer in captive environments such as zoos compared to in the wild, although some long-lived species such as elephants and chimpanzees do not (Tidière et al., 2016).

Table 1 Summary of existing studies describing wild or captive chimpanzee life history patterns.

Source	Population (captive or wild)	Location and study years	Sample size	Life expectancy of live born individuals	Life expectancy of individuals who survived beyond 1 year (calculated from life table)	
Havercamp et al., 2019	Captive	Japan
1921–2018	821	Both (sexes) = 28.3 y	From 1 y = 34.6 y
From 12 y = 40.4 y
From 15 y = 42.4 y	
Littleton, 2005	Captive	Australia
1941–2000	113	Not presented,
no life table	Not presented	
Dyke et al., 1995	Captive	A total of three breeding colonies in the United States
(years unknown)	F = 650
(Total N = 1,346, but ex not provided for males)	Females only (observed data) = 23.1 y, not reliable due to young population at the time	Not presented–only model life tables presented	
Courtenay & Santow, 1989	Captive	Australia & New Zealand
1935–1983	87	Not possible to calculate late life expectancy due to small, young population (mortality probabilities up to 30 y presented)	Not presented	
Wood et al., 2017	Wild	Ngogo, Uganda
1995–2016	306	Both = 32.8 y	From 1 y = 38.6 y
From 12 y = 43.1 y
From 15 y = 43.1 y	
Bronikowski et al., 2016	Wild	Gombe, Tanzania
1963–2013	F = 155
(Total N = 288, but ex not provided for males)	Females only = 15.9 y	Females only =
From 1 y = 19.4 y
From 12 y = 30.2 y
From 15 y = 32.0 y	
Bronikowski et al., 2011	Wild	Gombe, Tanzania
1963–2008	F = 144
M = 122	Females = 16 y (range: 10-25y)
Males = 11 y (range: 9-14 y)	Not presented	
Muller & Wrangham, 2014	Wild	Kanyawara, Uganda
1989–2013	123	Both = 19.4 y	From 1 y = 22 y
From 12 y = 34.5 y
From 15 y = 39.2 y	
Hill et al., 2001	Wild	5 study populations
Gombe, Tanzania: 1963–1998;
Taï, Ivory Coast: 1982–1994; Kanyawara, Uganda: 1989–1998;
Mahale, Tanzania: K group 1966–1988, M group 1979–1988;
Bossou, Guinea: 1976–1993	179 +
123 +
74 +
92 +
22
= 490	Both = 12.9 y	From 1 y = 15.6 y
From 12 y = 27.4 y
From 15 y = 29.8 y	
Note:

When full life tables are presented for both females and males, life expectancy estimates are combined.

There are several existing reports on captive chimpanzee survival which serve to inform our understanding of life histories (Table 1). Unfortunately, they are now outdated, derived from relatively small samples or missing data on older individuals. Courtenay & Santow (1989) published the first life table, though it was derived from just 87 individuals and mortality estimates were only calculated up to age 30 due to the short history of the population at that time. Dyke et al. (1995) presented modelled data including 1,488 individuals from three colonies in America and reported that males and females had an expected lifespan of 21 years and 29 years respectively when calculated from birth. When calculated from adulthood (14 years), males had an expected lifespan of 33 years and females 41 years. However, because the modelled life tables represented an average of various sources of variability, they urged caution in using these data as a representation of chimpanzees. In a later study, chimpanzees at Taronga Park Zoo in Australia experienced higher mortality risk from birth than wild chimpanzees in Mahale or Gombe. They also reported that females had greater life expectancy than males, however the size of the captive population was only 113 and the life table reached only 51 years of age (Littleton, 2005).

Historically, America and Japan have held the largest number of captive chimpanzees in the world, over 3,000 individuals combined, and the trajectory of those populations have been described in detail by Hirata et al. (2020). The earliest record of a chimpanzee in America is from 1902, whereas the first chimpanzee arrived in Japan around two decades later, in 1921. America adopted the Convention on International Trade in Endangered Species of Wild Fauna and Flora (CITES) in 1975, ceasing the importation of chimpanzees captured in Africa. Japan followed soon after ratifying CITES in 1980, however was still able to import wild chimpanzees under special circumstances for research until 1983. Broadly speaking, the populations are relatively similar in terms of their management at an individual and group level, however there are likely some differences at the population level. While the Japan population is largely self-contained and its growth is dependent on births in zoos, the America population has a significant proportion of chimpanzees that have entered the population from external sources such as research centers, the entertainment industry, and the pet trade (Hirata et al., 2020).

We synthesize data from these two regional captive populations to yield the largest dataset on chimpanzee life history and report the first systematic comparison of survival patterns for a primate species across regions. We present life tables by age and sex and statistics for chimpanzee survival and life expectancy for both the America population living in zoos accredited by the Association of Zoos and Aquariums (AZA) and the Japan population living in zoos and one sanctuary. We use studbook records, covering the modern history of these apes in each country to examine whether differences in longevity and survival exist between the two populations over a comparable time period. We hypothesized that the two populations would show similar life history patterns for survivorship and longevity, and demonstrate similar influences of sex and birth type (wild or captive born) on these variables. We did not have a priori predictions on how the survival patterns may differ between regions based on our knowledge of the welfare and management of the two populations. Factors contributing to survival rates such as diet, the ways and levels of veterinary care, and the specific physical and social environments differ between regions but also vary greatly between facilities within each region. Thus there is unlikely to be a consistent difference between countries that would impact survival metrics at the regional scale. We also predicted that survivorship for both populations would increase in the more recent timeframe (2001–2020) and tested whether mortality varied by season in the AZA population as was shown in the Japan population (Havercamp et al., 2019). Such cross-region population comparisons may help elucidate potential life history outcomes affected by differences in captive care and management.

Materials & methods

Data collection

We accessed demographic data for the chimpanzee population housed in AZA accredited facilities via the North American Regional Studbook for Chimpanzees (Ross, 2020), the most robust and consistent record of chimpanzees in the United States. Over more than a century, thousands of chimpanzees have lived in a range of captive settings in America including laboratories, zoos (accredited and unaccredited), sanctuaries (accredited and unaccredited), and in private hands (as pets or performers). Due to the varied nature of housing facilities and the historically loose regulation of cross-facility transfers, there was no comprehensive record of the U.S.-based chimpanzee population until Lincoln Park Zoo’s Project ChimpCARE, founded in 2007, began censusing all chimpanzees in the country (www.chimpcare.org). The North American Regional Studbook (Ross, 2020) contains records of 1302 chimpanzees since 1902, most of whom have lived in accredited zoos. At the time of writing, 260 chimpanzees lived in 31 AZA-accredited facilities in the United States, whereas Project ChimpCARE estimates that 1382 were alive across all settings (including zoos, sanctuaries, private owners, laboratories, and unaccredited facilities).

Similarly, we accessed data on captive chimpanzees living in zoos and a sanctuary in Japan via the open-access Great Ape Information Network (GAIN; https://shigen.nig.ac.jp/gain/; Havercamp et al., 2019) using PopLink software (Faust et al., 2019). Chimpanzees in Japan have been tracked via GAIN since the program was initiated in 2002 (Matsuzawa, 2016; Ochiai et al., 2015; Watanuki et al., 2014). The database has been utilized for a recent demographic analysis of this population (Havercamp et al., 2019). There are records for 1,025 (possibly 1,067) individuals who have lived in Japan since 1921, most of whom were housed in zoos, but also in eight biomedical research institutions, universities, and a sanctuary. In total, 205 were subjects of biomedical research (Hirata et al., 2020). At the time of writing, 302 chimpanzees live across 48 facilities including 46 zoos, animal or amusement parks, one university institute (a total of 12 individuals; Primate Research Institute of Kyoto University, Aichi, Japan), and one sanctuary (a total of 50 individuals; Kumamoto Sanctuary of Kyoto University, Kumamoto, Japan established in 2011, which was a former biomedical institution before it was reformed into Chimpanzee Sanctuary Uto, Kumamoto, Japan, in 2007; Morimura, Idani & Matsuzawa, 2011).

For both populations, we included for analysis individuals with a known birth location who had at least a portion of their lifespan within a time frame that reflected modern population management (Jan. 1, 1975 to Feb. 1, 2020 for AZA, Jan. 1, 1980 to Feb. 1, 2020 for Japan). We selected these ranges based on the years when America and Japan ratified CITES. For the AZA population, we also filtered to include the portion of lifespans that occurred within AZA-accredited facilities. We did not include records for chimpanzees who did not live in accredited zoo settings (i.e., laboratory, private, sanctuary). Some individuals had uncertainty in their birth dates, and our results should be interpreted with these in mind. Of the 730 total individuals (320 males, 389 females, and 21 individuals of unknown sex) across 94 facilities included from the AZA population, 563 had known birthdates. The remaining had a recorded birth date plus a time window reflecting the recorder’s uncertainty in the actual birth date (e.g., Jan. 1, 1993 ± 1 month): three individuals had estimates that were within 4 years of the actual birth date, 104 within 2 years, 28 within 1 year, one within 6 months, and 31 within 1 month. Of the 660 total individuals (292 males, 364 females, four individuals of unknown sex) across 77 facilities included from the Japan population, 486 had known birth dates, while 172 had birth date estimates that were within 1 year and two had estimates within 1 month of the actual birthdate. Finally, we excluded miscarriage and stillbirth events in the studbook (records of individuals who died on their birthdates), of which there were 59 in AZA and 50 in Japan.

Life tables

Using these data, we calculated life tables for each population using PMx software (Ballou, Lacy & Pollak, 2020), consisting of Kaplan–Meier estimates of annual age-specific mortality (qx) and fecundity (Mx) rates (Lacy, Ballou & Pollak, 2012). We included individuals of unknown sex, all of whom died (four individuals in Japan and 19 in AZA) or were lost to follow-up (two in AZA) within their first year of life, in the life table calculations to more accurately reflect the observed rate of first-year mortality. They were distributed as 0.5 male and 0.5 female following the standard in PMx (individuals of unknown sex were excluded from all remaining analyses). To compare against a recently published life table for a wild chimpanzee population in Gombe, Tanzania (Bronikowski et al., 2016), we calculated lx (survival to age x) using qx (probability of death within a year for age x individuals) from our PMx life tables as: lx = lx − 1 * (1 − qx − 1), and lo = 1.0.

Survival patterns and population comparisons

We conducted survival analyses (Kleinbaum & Klein, 2005) using the survival R package (Therneau, 2021) to describe the survival patterns in each population and to test for differences between the two populations. First, we used Cox proportional hazards regression to test whether survival curves differed by sex, birth type (wild-born or captive-born), and their interaction for each population separately. After excluding individuals of unknown sex, there were 69 wild-born males, 251 captive-born males, 100 wild-born females, and 289 captive-born females in the AZA population. For the Japan population, there were 62 wild-born males, 230 captive-born males, 105 wild-born females, and 259 captive-born females. We then combined the data for both populations and tested whether survival curves differed by sex, birth type, region (AZA or Japan), and all possible interactions. We removed any non-significant (P > 0.05, but all had P > 0.1) interaction terms in the final models. We also used the Kaplan–Meier method to fit observed survival curves based on the significant predictors, and to estimate the median life expectancy for each population by sex. We performed survival analyses starting at birth and starting at age one to assess whether patterns change when neonatal mortality was excluded.

Distribution of age at death by population

We next examined longevity over time in each population. We first calculated the age at mortality for each observed death event and divided the events into two time periods (“early” = 1975–2000 for AZA and 1980–2000 for Japan, “recent” = 2001–2020 for both). In total, there were 197 deaths observed in the early period and 150 in the recent period for the AZA population, and 139 deaths observed in the early period and 168 in the recent period for the Japan population. We then used Poisson regression to test whether the mean age at mortality differed based on sex, birth type, and between the two time periods. That is, we modeled the observed ages at mortality as a Poisson-distributed response variable in a generalized linear model with a log link function to analyze the effects of sex, birth type, and time period on the mean age at death. We used the function glm in the stats R package (R Core Team, 2020), and specified the quasipoisson family to account for overdispersion in the response. Because no deaths before age one were observed in wild born individuals, we conducted the analysis using only mortalities that occurred on or after the first birthday.

Seasonal death patterns

Finally, we assessed seasonal patterns in mortality for the AZA population, replicating an analysis previously conducted for the Japan population (Havercamp et al., 2019). For this, we tallied the number of observed deaths by season (spring = March through May, summer = June through August, autumn = September through November, winter = December through February). We applied a Chi-square goodness of fit test (using function chisq.test in the stats R package (R Core Team, 2020)) to compare the observed proportion of mortality in each season to the expected proportion (25% each, or evenly distributed across seasons). We also analyzed only mortalities that occurred on or after the first birthday, and re-analyzed data from the Japan population to match the time frame used in this study (1980–2020). Except where specified, all survival and statistical analyses were performed in R (R Core Team, 2020) and statistical significance was determined using the criterion P ≤ 0.05.

Results

Life tables

We present the full life table for the AZA population and Japan population (Tables S1 and S2, respectively; Fig. 1). The longest observed lifespan was longer in the AZA population, with one female surviving to an estimated age of 79, whereas one male survived to estimated age 68 in the Japan population. The longest-lived captive-born (confirmed age) individual in AZA was 57 and in Japan is 53 (still alive). First year mortality was significantly lower in AZA (15%) than Japan (25%) for females (z = −2.3, P = 0.02), and similar between AZA (18%) and Japan (20%) for males (z = −0.6, P = 0.58). Comparing the survival curves for our two captive populations against that for a wild population from Tanzania, we found that captive chimpanzees experienced a higher survival probability than wild chimpanzees starting at age 2 (Fig. 1).

Figure 1 Age-specific survival rates for females and males, from life tables for a wild population and the AZA and Japan captive populations of chimpanzees from birth (top) and from age 1 (bottom).

Wild population data is from Gombe, Tanzania (Bronikowski et al., 2016).

Survival patterns and population comparisons

When analyzing each population separately, we found survival patterns differed by sex but not birth type in the AZA population. AZA males had higher overall mortality rates than AZA females, both when examining survival from birth (coefficient = 0.41, z = 2.7, P < 0.01) and from age one (coefficient = 0.51, z = 2.8, P < 0.01). Survival patterns did not differ by sex or birth type in the Japan population (P > 0.15 in all cases). Accordingly, median life expectancy estimates from birth are lower for males than females in AZA (28.4 years and 36.7 years, respectively) but similar between sexes in Japan (34.0 years and 34.1 years, respectively; Table 2). From age one, a similar difference in median life expectancy exists between the sexes in AZA (32.5 years for males, 41.9 years for females), and a slight difference appears in Japan (37.7 years for males, 42.4 years for females; Table 2).

Table 2 Kaplan–Meier median life expectancy (MLE) estimates for the AZA population (1975–2020), the Japan population (1980–2020), and both populations of captive chimpanzees combined.

	Female MLE	Male MLE	
From birth			
AZA	36.7 (32.7–42.1)	28.4 (21.7–32.5)	
Japan	34.1 (30.2–41.2)	34.0 (27.1–38.0)	
AZA + Japan	35.7 (32.4–40.0)	30.1 (27.3–34.3)	
From age one			
AZA	41.9 (37.4–47.5)	32.5 (29.4–37.5)	
Japan	42.4 (39.9–47.2)	37.7 (34.8–NA)	
AZA + Japan	42.4 (40.0–46.3)	35.5 (32.6–38.0)	
Note:

Separate estimates are presented for females and males and starting from birth or from age one (i.e., assuming survival to first birthday), with their 95% confidence intervals in parentheses. For the Japan population male estimate, the upper limit was undefined because many older males are still living.

When combining data from both populations to compare survival patterns between regions, the results differed slightly depending on whether we included or excluded neonatal mortality (Fig. S1). Survival patterns from birth showed a significant effect of sex (coefficient = 0.39, z = 3.2, P < 0.01) and an interaction between region and sex (coefficient = −0.39, z = −2.3, P = 0.02): males have lower survival than females overall, but the difference is larger in AZA than Japan. Survival from age one showed an effect of sex (coefficient = 0.31, z = 3.0, P < 0.01), again with higher mortality for males. The Japan population had slightly lower mortality rates than the AZA population, but the difference was not significant (coefficient = −0.19, z = −1.8, P = 0.07; Fig. S1) and there were no significant interactions among predictors.

Distribution of age at death by population

Comparing longevity over time also showed slight differences between populations, with interacting effects of sex, birth type, and time period on age at death for both populations (Table 3; Fig. 2). Although we did not analyze deaths before age one in this analysis, we note that more neonatal deaths occurred before 2001 than since 2001 in captive-born individuals; neonatal deaths were not possible to observe in wild born individuals. For the AZA population, 43 of 141 observed deaths (30%) in the early time frame occurred before the first birthday, compared to 15 of 139 (11%) deaths observed in the recent time frame having occurred before age one. For the Japan population, 45 of 111 deaths (41%) in the early time frame occurred before the first birthday, compared to 43 of 146 deaths (29%) in the recent time frame.

Figure 2 Observed distribution of ages at death by sex, time period, and birth type in (A) the AZA and (B) the Japan captive chimpanzee populations.

Only deaths occurring on or after age one are included. The “early” time period is 1975–2000 for AZA and 1980–2000 for Japan, and the “recent” period is 2001–2020 for both. Sample sizes for the number of observed deaths for females and males are included in parentheses for each group. Note that the y-axis range differs between panels.

Table 3 Results from Poisson regressions comparing whether the median age at mortality in the AZA and Japan populations differed based on sex, birth type, and between two time periods in captive chimpanzees.

	Estimate	Std. Error	z value	p value		Estimate	Std. Error	z value	p value	
AZA					Japan					
(Intercept)	2.41	0.12	20.75	<2E − 16	(Intercept)	2.42	0.16	15.41	<2E − 16	
Sex (male)	−0.08	0.15	−0.53	0.5951	Sex (male)	0.46	0.22	2.08	0.0388	
Period (late)	0.89	0.13	6.82	9.33E − 11	Period (late)	0.68	0.18	3.83	0.0002	
Birth type (wild)	0.87	0.13	6.50	5.52E − 10	Birth type (wild)	0.71	0.19	3.83	0.0002	
Sex (male) * period (late)	−0.16	0.14	−1.18	0.2407	Sex (male) * period (late)	−0.43	0.25	−1.73	0.0863	
Period (late) * birth type (wild)	−0.32	0.14	−2.19	0.0297	Sex (male) * birth type (wild)	−0.73	0.26	−2.78	0.0061	
Sex (male) * birth type (wild)	0.13	0.13	0.97	0.3319	Period (late) * birth type (wild)	−0.15	0.21	−0.74	0.4631	
Null deviance: 2528.3 on 221 degrees of freedom	Sex * birth type * period	0.69	0.31	2.24	0.0263	
Residual deviance: 1255.8 on 215 degrees of freedom	Null deviance: 1203.96 on 168 degrees of freedom			
Dispersion parameter for quasipoisson family taken to be 5.782681	Residual deviance: 686.09 on 161 degrees of freedom			
					Dispersion parameter for quasipoisson family taken to be 4.176631	
Note:

The “early” time period is 1975–2000 for AZA and 1980–2000 for Japan, the “recent” period is 2001–2020 for both. Analyses were performed using only mortalities that occurred on or after the first birthdate. Bolded coefficient estimates highlight the effects that were significant at p < 0.05.

For the AZA population, the mean ages at death differed by time period and birth type but not by sex (Table 3), with mortality on average occurring later for wild born individuals and in the recent time frame (Fig. 2A). The two-way interaction between time period and birth type was also significant (Table 3), with a slightly larger difference in age at death between captive and wild born individuals in the later time frame compared to the earlier time frame (Fig. 2). When focusing specifically on the early time period for the AZA population, we found the median age of death was very similar for wild-born males and females (F = 26.4, M = 27.8) and lower for captive-born individuals (F = 11.1, M = 10.3). Considering only the more recent time frame, the mean age at death was higher for both wild-born (F = 47.0, M = 42.0) and captive born (F = 27.1, M = 21.3) chimpanzees. It is important to note that the population was younger overall in the early time period: the mean age of the AZA population was only 14.5 in the early time period (averaged across years within the time frame), and 23.7 in the recent time period (see Fig. 3 for mean age of the population in each year).

Figure 3 Median age and total population size of the AZA and Japan captive chimpanzee populations over time.

Ages were tabulated starting at age one and censused at Dec. 31 of each year (except for the last year, which was censused on the last data compilation date of Feb. 1, 2020). Individuals who had not yet reached age one or had an unknown birth location (30 individuals in the Japan population and 6 in the AZA population as of Feb. 2020) were not included in this figure.

For the Japan population, the mean ages at death differed by all predictors (Table 3), and the three-way interaction as well as the two-way interaction between sex and birth type were also significant (Table 3). The three-way interaction meant that, for example, the difference in age at death between males and females depended on both the time period and birth type. Captive born females died at a younger age than captive born males, but wild born females survived to an older age than wild born males, and this was more apparent in the early time period. The estimated mean age at death in the early time period was 22.9 and 17.5 for wild born females and males, respectively, and 11.3 and 17.8 for captive females and males, respectively. In the recent time period, the estimated mean age at death was 38.7 and 38.4 for wild born females and males, respectively, and 22.2 and 22.9 for captive females and males, respectively. For context, the mean age of the Japan population was 11.1 in the early time period and 22.2 in the recent time period (Fig. 3).

Seasonal death patterns

In examining seasonal patterns of mortality, we confirmed previously published findings (Havercamp et al., 2019) that deaths in the Japan population were not evenly distributed across seasons. To compare to these earlier reports, we used only data since 1980 and found a significant deviation from the expectation of even distribution both when considering all deaths (χ2 = 13.77, df = 3, P < 0.01) and when excluding neonatal mortality (χ2 = 11.95, df = 3, P < 0.01), with the largest proportion of deaths occurring in winter. We did not find such a seasonal pattern in the AZA population, both in analysis of all deaths (χ2 = 6.37, df = 3, P = 0.09) and when considering only deaths after age one (χ2 = 4.20, df = 3, P = 0.24; Fig. 4).

Figure 4 Numbers of deaths in the AZA (top panel) and the Japan (bottom panel) captive chimpanzee populations by month (AZA: 1975–2020, Japan: 1980–2020).

Seasons are grouped in alternating colors: winter = Dec–Feb (green); spring = Mar–May (gray); summer = Jun–Aug (green); Fall = Sep–Nov (gray). Darker bars include all observed deaths, and lighter bars include only deaths on or after age one.

Discussion

Although there are several published reports on life histories of captive chimpanzee populations (Courtenay & Santow, 1989; Dyke et al., 1995; Littleton, 2005; Havercamp et al., 2019), we believe this is the first to compare the life histories of two large captive chimpanzee populations from different regions. Such a comparative approach provides additional insights that may inform managers about the life history characteristics that are more (or less) flexible or potentially affected by different management practices.

Both of these populations of chimpanzees have been evaluated in previous studies, but the current datasets differ from previous ones in some important ways, primarily to facilitate cross-region comparisons. Earnhardt et al. (2003) used data from the AZA studbook to compare life histories of these chimpanzees to wild chimpanzees living at Gombe Stream National Park in Tanzania. That dataset represented 524 zoo-housed chimpanzees living from 1963 to 2003, compared to 730 from 1975 to 2020 in the current analysis. Likewise, an earlier study (Havercamp et al., 2019) analyzed the historical chimpanzee population living in Japan, using data for 821 chimpanzees from 1921 to 2018 compared to 660 from 1980 to 2020. For both populations, the current analysis represents a more narrow and more recent time frame, and the median life expectancies from the previous analyses were lower than those derived from this study, especially for females. This may be attributed in part to the more recent time frame reflecting more modern management practices that positively influence life history outcomes for these managed populations. For the AZA population, the earlier analysis (Earnhardt et al., 2003) included data up to 2003 and thus the younger population at that time (Fig. 3) could have also contributed to the shorter life expectancy estimates. However, the earlier analysis of the Japan population (Havercamp et al., 2019) included data through 2018 so the difference in our analysis is mainly excluding the earlier records, and thus are more likely to reflect recent refinements in management.

Combined, the dataset used here represents the lives of 1390 chimpanzees living in accredited zoos in America and zoos and a sanctuary in Japan. Using data from birth, median life expectancy (MLE) is 35.7 for females and 30.1 for males (Table 2), which is higher than most longevity estimates for wild chimpanzees. Hill et al. (2001) reported an MLE of just 13 years from data collected at five sites across Africa (see Table 1), and Muller & Wrangham (2014) reported an MLE of 19 years at Kanyawara, Uganda. However, recent data from Ngogo, Uganda placed life expectancy at over 33 years for that population (Wood et al., 2017). In a direct comparison of wild and captive chimpanzee survivorship, Earnhardt et al. (2003) found that chimpanzees from AZA zoos typically lived longer than those living in Gombe Stream National Park in Tanzania. In comparing the survival curves of our two captive populations with a wild chimpanzee population living in Tanzania (Bronikowski et al., 2016; Fig. 1), we also found support for this pattern; at almost all ages from birth and from 1 year of age, captive chimpanzees experienced a higher survival probability. Interestingly, although Tidière et al. (2016) showed that mammals generally live longer in captive environments such as zoos compared to in the wild, they found the opposite trend for chimpanzees specifically. Moreover, survival statistics can vary greatly across wild populations (Davison & Gurven, 2021), so our comparison does not represent all wild populations (i.e., see Wood et al., 2017 for a population with higher survivorship). While these comparisons are worthwhile, we must acknowledge that survivorship comparisons are complicated because of the difficulty in assessing early life mortality for wild chimpanzees, including stillbirths. We removed stillbirths from these analyses, but it is impossible to say how stillbirth rates in captive settings, documented at 12% for AZA (Saiyed et al., 2018) and 15% for Japan (Havercamp et al., 2019), compare to those experienced by wild-living chimpanzees.

One of the most robust findings in this analysis was the sex difference in mortality demonstrated in the AZA chimpanzee population. Whether calculated from birth or after the age of one, males had higher mortality rates and shorter life expectancies when compared to females in the same managed population. Such sex differences are reflected by findings from a number of wild chimpanzee populations (Hill et al., 2001; Muller & Wrangham, 2014; Wood et al., 2017) as well as other mammals (Tidière et al., 2016) and are broadly consistent with sexual selection theory. For many species, including chimpanzees, males are perceived as engaging in more risky behaviors (for instance, hunting and intergroup aggression) which increase the mortality risk. The human literature also reports female advantages in life expectancy that could be tied to a range of genetic, hormonal, metabolic, immune function and other biological factors (Seifarth, McGowan & Milne, 2012). For instance, Vina et al. (2011) showed that the higher levels of estrogens in human females buffer them against the negative effects of aging, by up-regulating the expression of antioxidant and longevity-related genes.

Interestingly, such sex differences were not evident in the Japanese population and may be explained in part by regional differences in early life histories. Female first-year mortality in Japan was significantly higher than in AZA, suggesting that challenges in birth management and early life care may contribute to increased female mortality rates and a narrowing of the sex gap in mortality one might expect. Comparing the two regions, while female survival seems similar in both regions, males in the AZA population have lower survival than males in the Japan population. Explanations for these differences are unclear to us at this time. Though housing conditions and group compositions are not recorded as part of these datasets, we know anecdotally that in general, chimpanzee groups in AZA are larger than those in Japanese zoos and are more likely to have a multi-male composition. Historically, some have speculated that males living with fewer or no other males may be under lower daily stress (Alford et al., 1995; Williams et al., 2010), but recent studies contradict this idea and demonstrate that males can live together in mixed-sex or multi-male groups without experiencing heightened aggression or stress levels (Neal Webb, Hau & Schapiro, 2019; Ross et al., 2009; Seres, Aureli & de Waal, 2001; Yamanashi et al., 2016). In a multi-institutional study of AZA-accredited zoos, multi-male groups showed lower rates of wounding than single male groups (Ross et al., 2009). Likewise, an all-male group did not show heightened hair cortisol (HC) concentration (a proxy for stress), whereas the alpha male in a mixed-sex group showed the highest level of HC and aggression (Yamanashi et al., 2016). Nonetheless, the effect of social group composition on health and stress levels is not clear, and so whether this may create a difference in male survival between the two regions is unknown. Other potential factors, such as diet differences in the two regions, may be particularly ripe for future study and we encourage further refinement and extension of cross-regional comparisons of life histories for chimpanzees and other species under intense population management.

In addition to sex effects on life expectancy, we examined the effects of whether chimpanzees were born in the wild or born in captivity. Overall survival patterns did not differ for those born in captive settings compared to those born in the wild, for either the AZA or the Japan population. However, differences in age at death between sexes depended on birth type. In the AZA population, males died younger than females, but this sex difference was smaller for wild-born vs. captive born individuals. In Japan, captive-born males tended to die later than captive-born females, but wild-born males died younger than wild-born females. These analyses of birth origin need to be interpreted in context of the fact that early life histories of wild-born individuals are not available (i.e., there are fewer observed deaths in the younger ages, before transfer into captive populations). Also, many wild-born individuals who were captured may not have survived the travails of capture and transport and are therefore not represented in these analyses. This may be an example of the related concepts of selective disappearance and mortality selection (Vaupel, Manton & Stallard, 1979; Vaupel & Yashin, 1985; Hämäläinen et al., 2014) in which the capture process could have selected for healthier individuals who would thus have higher survival going forward. This may also explain why we did not see the effect of birth type in the survival analyses, which take into account which portions of the lifespan are observed and therefore provide more robust results. As such, we urge caution against over-interpretation of the results that may suggest that wild-born chimpanzees have some form of extra resilience that boosts their longevity.

We also examined how life history characteristics may have changed over time by comparing data from an early period (1975/1980–2000) to a later period (2001–2020). In both regions, age at death has increased in the most recent 20-year period compared to the two decades prior. However, it is important to note that this only represents deaths that had occurred in these periods and because the population was younger overall in the early period, the average age at death was also younger. For example, it does not mean that a captive-born AZA male living in the early time period would only live to ten; rather that captive-born AZA males that died in the early time period were on average aged ten. The median age of both populations is increasing over time (Fig. 3) and will likely continue to increase as more animals have the chance to live out their lifespan. These results demonstrate that focusing only on ages at death can be misleading for determining animal life expectancy, similar to an earlier study on captive elephants (Wiese & Willis, 2004). This artefact also makes it difficult to test whether there are differences in survival over time due to advancements in care and management protocols as well as veterinary capabilities over the past 40 years. In the AZA population for instance, there have been significant improvements in a number of aspects of behavioral management including the promotion of larger and more natural (e.g., multi-male) groups and the integration of fission-fusion management systems that may promote a more natural social setting for zoo-housed chimpanzees. Likewise, veterinary expertise surrounding sedation protocols (Naples, Langan & Kearns, 2010), diet (Struck et al., 2007) and wound management (Baker et al., 2000) may contribute to improving health, welfare and potentially longevity in recent years. Similar activities are being done in Japan as well to promote environmental enrichment (Morimura & Ueno, 1999; Morimura, 2003), stress monitoring (Yamanashi et al., 2013), method of sedation (Miyabe-Nishiwaki et al., 2021), and detection of zoonotic pathogens (Kooriyama et al., 2013).

Finally, we validated the findings of Havercamp et al. (2019) showing that chimpanzees in the Japan population were more likely to die in winter months. For the AZA population, our data suggest that deaths may peak in both the winter and summer periods (compared to spring and autumn; Fig. 4) but this seasonal pattern was not statistically significant. Although Japan is geographically much smaller than America, both countries are characterized by a broad degree of climatic variation. We are unable to specify if such impacts are particularly prevalent in regions which experience particularly cold winters, though further detailed study of these climate-related effects would be welcome. Nonetheless, these findings provide further support that husbandry efforts should be intensified during the winter season (e.g., continuous availability of temperature controlled rooms, special diet, warming enrichment, additional health monitoring), especially towards infants, as chimpanzees are likely not well adapted to cold weather and its effects on health. The challenges of managing chimpanzees, who have evolved in equatorial climates, in winter times remains an important consideration.

Conclusions

We compiled the largest dataset on chimpanzee life history, consisting of nearly 1400 individuals and spanning four decades, and compared life history patterns in captive chimpanzee populations from two different regions. Despite some differences in management practices, the overall life expectancy and survival patterns were similar between the two regions. However, survivorship was lower for males than for females in the AZA population but similar among sexes in the Japan population. The Kaplan-Meier median life expectancy from birth was 35.7 (32.4–40.0) years for females and 30.1 (27.3–34.3) years for males across both populations, which is higher than most longevity estimates reported for wild chimpanzees. Birth type (wild-born or captive-born) did not influence survival patterns in either population, and the seasonal death pattern previously shown in the Japan population was not found in the AZA population. We were unable to fully assess changes in survival over time due to the long lifespan of the species and many individuals having yet to live out their full lives. Moreover, our current estimates of median life expectancy will likely increase in future investigations as animals continue to age in both populations. The opportunity to quantitatively compare life history patterns between large populations of managed species is relatively rare, especially with large and long-lived animals such as chimpanzees. Further investigation will increase the potential of these data to inform important population management strategies.

Supplemental Information

Supplemental Information 1 Modeled survival curves based on Cox proportional hazards model for females and males in AZA (1975–2020) and Japan (1980–2020) captive chimpanzee populations, starting from birth or from age one.

Confidence intervals not shown as they overlap and obscure patterns. Note that the model for survival curves from birth (left) included the interaction between region and sex, whereas the model from age one (right) included no interaction effects. Because there was no significant difference between the sexes in survival from birth for the Japan population, the Japan male and female (red and purple) lines are overlapping in the left panel.

Click here for additional data file.

Supplemental Information 2 AZA captive chimpanzee life table (1975–2020).

In these life tables (Tables S1 and S2), nx (qx) = number of individuals at risk for mortality calculations, dx = probability of death between age x and x + 1 calculated as l(x + 1) − l(x), qx = probability of death between age x and x + 1 calculated as the number of animals that die during an age class divided by the number of animals at risk, lx (0y) = probability of survival from birth to age x, lx (1y) = probability of survival from 1 year to age x, ex = remaining life expectancy (in years) at age x, nx (mx) = number of individuals at risk for fecundity calculations, mx = fecundity or the average number of same-sex young born to individuals in that age class.

Click here for additional data file.

Supplemental Information 3 Japan captive chimpanzee life table (1980–2020).

Please see Table S1 legend for definitions of abbreviated terms.

Click here for additional data file.

We thank Lisa Faust and Kristine Schad Eebes for helpful discussions regarding studbook analyses. We are also grateful to Yasuhiro Yoshikawa, Toshikazu Hasegawa, Gen’ichi Idani and all members of the Great Ape Information Network for their support in data collection in Japan.

Additional Information and Declarations

Competing Interests

Author Contributions

Data Availability

The authors declare that they have no competing interests.

Judy Che–Castaldo conceived and designed the experiments, performed the experiments, analyzed the data, prepared figures and/or tables, authored or reviewed drafts of the paper, and approved the final draft.

Kristin Havercamp conceived and designed the experiments, performed the experiments, prepared figures and/or tables, authored or reviewed drafts of the paper, and approved the final draft.

Koshiro Watanuki conceived and designed the experiments, authored or reviewed drafts of the paper, and approved the final draft.

Tetsuro Matsuzawa conceived and designed the experiments, authored or reviewed drafts of the paper, and approved the final draft.

Satoshi Hirata conceived and designed the experiments, authored or reviewed drafts of the paper, and approved the final draft.

Stephen R. Ross conceived and designed the experiments, authored or reviewed drafts of the paper, and approved the final draft.

The following information was supplied regarding data availability:

Anonymized, individual-level data and R script for all analyses and figures are available at Figshare: Che–Castaldo, Judy (2021): Comparative survival analyses among captive chimpanzees (Pan troglodytes) in America and Japan. figshare. Dataset. DOI 10.6084/m9.figshare.14685429.v1.

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
