# Peer review of "Comparative survival analyses among captive chimpanzees (Pan troglodytes) in America and Japan"

_PeerJ, doi:10.7717/peerj.11913_

## Round 0.1 · original submission · Major Revisions

Dear Dr. Che-Castaldo and colleagues:

Thanks for submitting your manuscript to PeerJ. I have now received three independent reviews of your work, and as you will see, the reviewers raised some concerns about the manuscript. Despite this, these reviewers are optimistic about your work and the potential impact it will have on research studying behavior of captive chimpanzees. Thus, I encourage you to revise your manuscript, accordingly, taking into account all of the concerns raised by both reviewers.

Please improve the presentation, clarity and organization of your manuscript (particularly the Introduction). Make sure the proper and most relevant references are cited. Add structure to the work by using headings and sub-headings where appropriate.

Please also ensure that your figures and tables contain all of the information that is necessary to support your findings and observations. Revise incorrect information. The Materials and Methods appear to be missing important information. All statistical methods should be adequately described such that they are repeatable. All of other methodologies must also be explicitly described such that they are also repeatable.

Please note that Reviewer 1 kindly provided a marked-up version of your manuscript.

Please address all other concerns of the reviewers.

I look forward to seeing your revision, and thanks again for submitting your work to PeerJ.

Good luck with your revision,

Best,

-joe

Reviewer 1 ·

Basic reporting

The English was clear and professional.

In the Introduction there was extensive background source citation and context, and in the Discussion there was thoughtful interpretation and comparison with previous studies.

Figures and tables were helpful. Suggest larger font size in Fig 1 and 2 and population labels (AZA, Japan) in Fig 4.

Yes, clearly defined study with coherent results.

Experimental design

Appears within the Aims and Scope of PeerJ.

Well-defined question, relevant and meaningful to primatologists, anthropologists, demographers and, zoro managers and aging researchers. Contributes to our understanding of captive chimpanzee mortality.

Apparently technically sound, though I am not expert in all the statistics employed. No apparent ethical concerns.

Methos appear sufficient and detailed enough to replicate the study.

Validity of the findings

Individual-level data does not accompany study but the age-specific mortality rates generated by their analysis is provided for the AZA and Japan groups of zoo data.

Data appear robust and sound, having been acquired from accredited zoos.

Conclusions well-stated and possible reasons behind the patterns they observed are well-treated in the discussion, anticipating many of the questions this reader had while reading the results.

Speculation was avoided, except for tentative explanations for why certain patterns existed.

Annotated reviews are not available for download in order to protect the identity of reviewers who chose to remain anonymous.

·

Basic reporting

The manuscript is correctly written and easy to read, but a little bit of structure would help. The literature references are completely adequate, but the Introduction lack of real hypothesis and predictions based on theoretical framework. Because of the high number of results presented, it is easy to get lost. The Results section would win in clarity with headings explaining the different subsection such as for the study of the general survival patterns, and then the distribution of age at death. These headings should be used in the M&M, Results and Discussion parts. Moreover, it is not clear for me what this study provides as new results. All the analyses presented here are very interesting but as no hypothesis nor predictions are presented in Introduction, it is not easy to understand if the authors try to answer to a specific question.

Experimental design

Introduction must be strengthened with clear hypotheses and predictions. Because of that, the research question seems to not fill any knowledge gap. Why comparing AZA and Japan data? Males and females? Wild- and captive-born individual survival parameters? What is the purpose of this study?
The Methods are well explained but some structure would help to better understand.

Validity of the findings

The Discussion is clearly the strength of this manuscript. Results are clearly discussed, and the authors provide interesting interpretations.

Additional comments

Abstract
Do you expect any environmental difference between these two populations? If yes which one and what impact on survival metrics?
Lines 37-39: What are the number between brackets? The range? The 95%CI? It should be precise.

Introduction
Line 76: some species such as elephant yes but more generally long-live species … such as chimpanzee according to Figure 2 from Tidière et al. 2016.
Lines 93-126: Although this information is required, I suggest putting it in the M&M and use the free space to describe more the hypothesis and the prediction expected. What is the interest to compare Japan and America data? What are you expecting from the sex differences?
Lines 134-136: Ok but why?

Material and methods
Line 167: You do not study fecundity here so why introducing it?
Lines 168-169: You have a remarkable high-quality dataset, why introducing error in the estimations attributing unsexed individuals half to males and half to females instead of removing these 4 individuals from the analysis?
Line 179-180: Based on which criteria?

Results
As said in the general comments, the Results would win in clarity with sub-headings introducing the different analyses done.

Discussion
Lines 317-319: This sentence is the same sentence used in Introduction, what is the new point?

Figure 1
Why the wild population chosen is Gombe? As you present different studies in Table 1 maybe you can explain why you chose the Gombe population to compare and not another one?
I found the Figure 1 not very informative as we cannot compare the survival patterns of males and females on a same site. Having clear hypotheses and prediction would help to provide adapted figures that show the answer to the hypothesis.

Figure 4
Rather than indicating in the legend the season maybe it could be easier to visualize if indicated in the figure directly?

Table 1
I am not sure to understand the criteria explaining which studies can be in this table and which cannot. For instance, why Bronikowski et al 2016 and Bronikowski et al 2011 are not in the table 1?

Table 3
According to this table there is difference between sexes, between periods and between wild- and captive born in the two datasets.
The problem of this table is that it gives comparison between females in early period and captive-born, and all others combinations. It is not easy to compare the two male populations for instance (captive- vs wild-born or early vs late period).

Reviewer 3 ·

Basic reporting

This paper uses an impressive dataset for such a long lived animal to give a clear overview of life history patterns in two captive populations. Generally the merits of the study are well explained and justified as well as honestly outlining the limitations. The results of the study are clearly explained, including some very complicated interactions. However, the authors should provide clearer explanations and clarifications in a few areas such as in describing the study population and the statistics used. Although the raw data of the life tables were provided, data needed for other analyses seem to be lacking and the authors should consider uploading R script code as well as data for full reproducibility.

For example, the description of the two study populations in the introduction (lines 102-113) needs to be clearer, more similar for example to how they are summarised in the discussion (lines 286-303). A couple of things which are not clear at the moment:
1) Are the 1302 chimps (line 110) historic and alive? If so, from what time frame?
2) There are 260 chimpanzees alive in AZA zoos, and 1395 were alive from all settings, is this latter number currently alive or in total? This is also a bit misleading as it sounds like the study will include both the AZA chimps and those in other environments, whereas it actually only includes those in AZA environments- you should either remove mention of the other institutions or make it clear that you are only studying the AZA subset.
This section could generally be made clearer by simplifying and structuring the information, for example:
"A total of n chimpanzees have been recorded in the USA between the years nnnn-nnnn. n have lived in accredited zoos, recorded into the North American Regional Studbook (nnnn-nnnn), and n in other settings, recorded by the Lincoln Park zoo project (nnnn-nnnn). This study includes only the n chimpanzees from AZA accredited zoos...." It is not clear at the moment from the introduction which chimpanzees mentioned were included in the study and the numbers given in the introduction (1302 line 110/ 1395 line 113) do not seem to correspond to the total of 1390 given in the methods/discussion (line 305).

The methods should also include a breakdown of the sample sizes for relevant variables across the two populations: e.g. alive/dead animals, captive/wild born, numbers in different institutions, early/recent time periods etc. In general, the methods would also benefit from being split into more sections for structure, eg. data selection, life tables, statistical analyses etc.

Abstract: It would help readers interpret the study to give sample sizes already in the abstract, as one of the advantages to the study is the number of individuals involved.

Line 139: Please give the reference to this previous study- Havercamp et al 2019?

Figure S1: This figure should have a legend explaining what the left and right figures are showing, and the line for Japan Captive F in the left plot is not visible (is it hidden under another line?).

There were a few cases of sentences which could be written more clearly and generally the manuscript could use a thorough proofread with some specific examples given here:

Line 65 should be “a captive population”
Line 105 should this be in the past tense- “there was”?
Line 128- It would be informative to give the sample size of the dataset here.
Line 131- the explanation of the AZA abbreviation should be given at this first mention (currently given later at line 144)
Line 132- If “American population” is used, should it be “Japanese population” rather than "Japan population" for consistency? Same for all other instances throughout the paper
Line 168- “unknown sex individuals” should be reworded to “individuals of unknown sex”
Line 175 “differences among” should be “differences between”
Line 191 “birthdate” should be “birthday”. Same for line 199 and other instances.
Line 205- needs rewording “surviving to estimated age 79” doesn’t sound right - "an estimated age of 79"?
Line 223- be consistent with the tenses, if it is “differed” should it also be “included/excluded”
Line 250: should have “chimpanzees” or “individuals” after captive-born
Table 1: Be consistent with using abbreviation for life expectancy (LE)- only used in final column and not necessary in that column as it is the column title. Also in Hill et al 2001: "Mahale: K group 1966 (M group 1979)-1988" it is not clear what K and M refer to and are the parantheses necessary?
Table 2 legend- "AZA SSP"- this is the first mention as far as I can tell of SSP -what does it stand for and is it necessary?
Figure 3 legend “data currentness date” should be reworded
Figure 4 key labels could be improved by including a space in "AllDeaths" and changing "DeathAfter1" to "Deaths after age 1".
Table 3- could estimate column in japan section be changed to one line? Please be consistent in how you write the p value (e-n or E-n).
The sentence line 308-311 is unclear, please rephrase it – five sites where? Is the Ngogo chimp population distinct from the Kanyawara population?
Line 343- should be “explained by” rather than “explained to”
Line 374 “in context to the fact” should be “in context of the fact”
Line 424- missing “for males” after 30.1

Experimental design

The methods require more information on the packages used for each statistical analyses including citations (lines 174/188/197), and how statistical significance was determined for each. The results were generally very clear with the complicated interactions well explained.

Line 166 onwards- Although a lot of these abbreviations are standard for life tables, and many have been defined, it could aid a general audience to explain all the terms in the life tables (Tables S1&S2), either in the text in this paragraph, or within the Tables.

Line 168 Please explain why individuals with unknown sex were retained and assigned 50/50 to males and females, rather than excluding these individuals as would be considered more standard protocol. It should also be considered that if you include imputed sexes for these individuals, it would be more appropriate to base them on the sex ratio of your sample, which is female biased, rather than 50:50.

Line 196- Please give some information about how the 4 seasons compare in timing and severity between the US and Japan.

Line 188 please specify what is meant by “poisson regression” – what type of model is this, a generalised linear regression model? If so, explain this and cite the package used. More information is also needed about the response variable- is it a count of individuals that have died at each age? Similarly when reporting the results of this model (line 232 onwards), no test statistics are given for different terms, only those in the table. The authors should consider using likelihood ratio tests or similar to assess the significance of terms (comparing models with and without the term of interest), rather than simply reporting the model output.

Line 242- This isn’t necessarily the case – when there are significant interactions, then it changes how you should interpret the main effects – actually all 3 predictors have an effect they are just all in interaction with eachother, same for line 257.

Lines 315 “we also found support for this pattern” – Although you explained this comparison in the methods, it is not mentioned in the results and it seems strange to mention it for the first time in the discussion - this should be included in the results when referring to Fig 1.

Line 376- Further to this, selective disappearance could apply here and it would be an interesting point to discuss; the capture process could have selected for only the “good quality” individuals to survive.

Validity of the findings

The authors do a very good job of explaining the limitations of their study as well as the merits. Paragraph lines 342-365 for example is a very interesting discussion of their findings and 381-403 gives an honest overview of the limitations to the dataset.

Paragraph lines 405-417: For the interpretation of this paragraph, it would be useful to know the number of different institutions sampled from across the two different regions (suggested in an earlier comment for the methods section)- would it be possible to look at different regions within the two different countries?

The underlying data have been provided for the life tables, although not all corresponding data is included for all the analysis to be replicated as far as I can tell (for example, information on time period collected/ birth origin/ season). The authors should also consider providing the R code associated with the analyses to ensure full reproducibility.

---

## Round 0.2 · accepted · Accept

Dear Dr. Che-Castaldo and colleagues:

Thanks for revising your manuscript based on the concerns raised by the reviewers. I now believe that your manuscript is suitable for publication. Congratulations! I look forward to seeing this work in print, and I anticipate it being an important resource for groups studying behavior of captive chimpanzees. Thanks again for choosing PeerJ to publish such important work.

Best,

-joe